



Organic matter cycling along geochemical, geomorphic and disturbance gradients in forests and
2                    cropland of the African Tropics - Project TropSOC Database Version 1.0

Sebastian Doetterl[1,2], Rodrigue K. Asifiwe[6], Geert Baert[3], Fernando Bamba[6], Marijn Bauters[3,4],
Pascal Boeckx[3], Benjamin Bukombe[2], Georg Cadisch[5], Matthew Cooper[8], Landry N. Cizungu[6],
Alison Hoyt[7], Clovis Kabaseke[8], Karsten Kalbitz[9], Laurent Kidinda[9], Annina Maier[1], Moritz
Mainka[2], Julia Mayrock[2], Daniel Muhindo[6], Basile B. Mujinya[10], Serge M. Mukotanyi[6], Leon
Nabahungu[12], Mario Reichenbach[2], Boris Rewald[11], Johan Six[1], Anna Stegmann[2], Laura
Summerauer[1], Robin Unseld[2], Bernard Vanlauwe[12], Kristof Van Oost[13], Kris Verheyen[4], Cordula
Vogel[9], Florian Wilken[1,2], Peter Fiener[2]
[1] Department of Environmental System Sciences, ETH Zurich, 8092 Zürich, Switzerland
[2] Institute of Geography, Augsburg University, Augsburg, Germany
[3] Department of Green Chemistry and Technology, Ghent University, Ghent, Belgium
[4] Department of Environment Forest & Nature Lab, Ghent University, Ghent, Belgium
[5] Institute of Plant Production and Agroecology in the Tropics and Subtropics, University of Hohenheim, Stuttgart,
Germany
[6] Faculty of Agricultural Sciences, Université Catholique de Bukavu, DR Congo
[7] Max Planck Institute for Biogeochemistry, Jena, Germany
[8] School of Agriculture and Environmental sciences, Mountains of the Moon University, Fort Portal, Uganda
[9] Chair of Soil Resources and Land Use, Institute of Soil Science and Site Ecology, TU Dresden, Germany
[10] Biogeochemistry and ecology of tropical soils and ecosystems, University of Lubumbashi, DR Congo
[11] Department of Forest and Soil Sciences, University of Natural Resources and Life Sciences (BOKU), Vienna, Austria
[12] Institute of Tropical Agriculture, Central Africa and Natural Resource Management, CGIAR, Nairobi, Kenya
[13] Earth and Life Institute,UCLouvain, Louvain-la-Neuve, Belgium.
Corresponding author: sdoetterl@usys.ethz.ch



**Abstract**
The African Tropics are hotspots of modern-day land-use change and are, at the same time, of
great relevance for the cycling of carbon (C) and nutrients between plants, soils and the
atmosphere. However, the consequences of land conversion on biogeochemical cycles are still
largely unknown as they are not studied in a landscape context that defines the geomorphic,
geochemically and pedological framework in which biological processes take place. Thus, the
response of tropical soils to disturbance by erosion and land conversion is one of the great
uncertainties in assessing the carrying capacity of tropical landscapes to grow food for future
generations and in predicting greenhouse gas fluxes (GHG) from soils to the atmosphere and,
hence, future earth system dynamics.
Here, we describe version 1.0 of an open access database created as part of the project
**"Tropical soil organic carbon dynamics along erosional disturbance gradients in relation
to variability in soil geochemistry and land use" (TropSOC).** TropSOC v1.0 contains spatial
and temporal explicit data on soil, vegetation, environmental properties and land management
collected from 136 pristine tropical forest and cropland plots between 2017 and 2020 as part of
several monitoring and sampling campaigns in the Eastern Congo Basin and the East African Rift
Valley System. The results of several laboratory experiments focusing on soil microbial activity,
C cycling and C stabilization in soils complement the dataset to deliver one of the first landscape
scale datasets to study the linkages and feedbacks between geology, geomorphology and
pedogenesis as controls on biogeochemical cycles in a variety of natural and managed systems
in the African Tropics.
The hierarchical and interdisciplinary structure of the TropSOC database allows for linking a wide
range of parameters and observations on soil and vegetation dynamics along with other
supporting information that may also be measured at one or more levels of the hierarchy.
TropSOC's data marks a significant contribution to improve our understanding of the fate of
biogeochemical cycles in dynamic and diverse tropical African (agro-)ecosystems. TropSOC v1.0
can be accessed through the supplementary material provided as part of this manuscript or as a
separate download via the websites of the Congo Biogeochemistry observatory and the GFZ data
repository where version updates to the database will be provided as the project develops.




## 1. Rationale to project TropSOC

### 1.1 Changing tropical environments in Africa

Tropical ecosystems provide many services of global importance. Tropical forests are among the largest terrestrial carbon (C) reservoirs and show some of the highest levels of biodiversity (Losos and Leigh, 2004; Pan et al., 2014). At the same time, tropical landscapes are among the most dynamic regions worldwide and hotspots of modern day land-use change (Hansen et al., 2013) as they have to provide food for some of the poorest yet fastest growing populations on the planet. In particular, the African continent is facing huge environmental and societal challenges with a projected population growth of 400% by the end of this century (Gerland et al., 2014), much of it happening in (sub-)tropical sub-Saharan Africa. In consequence, forested landscapes in tropical Africa are currently facing unprecedented levels of land conversion and land degradation, accompanied by decreasing soil fertility (UNESCO and WHC, 2010). At the same time, unlike other tropical regions of the world, where deforestation are driven by the extension of commodity plantations and commercial logging, much of the deforestation in tropical African countries is driven by smallholder farms that apply slash and burn practices for subsistence farming with little alternatives to provide food for their families (Curtis et al., 2018; Tyukavina et al., 2018). As a result, deforestation and soil degradation have accelerated greatly since the second half of the 20[th] century with soil erosion, in particular, emerging as the main driver of soil degradation.

Today, erosion rates of tropical agricultural land globally are estimated at approx. 10.4 billion tons of soil per year and 0.2 billion tons of C per year. Tropical agricultural soil erosion represents therefore about half of the annual agricultural erosion globally, while only representing about one third of global cropland (Doetterl et al., 2012). An exemplary region to observe the consequences of land use change on soil resources and biogeochemical cycles in the tropical African region context is the African great lakes region along the East African Rift Valley System along the borders between the Democratic Republic of the Congo, Burundi, Rwanda and Uganda.

The region is a model for the complex interplay of socio-economic factors and their consequences for environmental systems in the Tropics. One of the highest human fertility rates globally (e.g. recent estimates for the last decade range from 7.3-7.7 children per woman in the province of South Kivu, Eastern DRC) (Dumbaugh et al., 2018) leads to massive population growth in the region, largely relying on local food and energy resources. Ridden by conflict and open warfare in the 1990s and early 2000s, population growth in the region is further aggravated



as a result of refugees from remote areas settling nearby safer, larger cities in the region
(Kuijrakginia et al., 2010). In consequence, massive deforestation of upland forests for fuel
gathering and cropland expansion is taking place (Hansen et al., 2013), leading to large erosional
soil fluxes and consequential soil degradation threatening soil quality (Karamage et al., 2016).
Once conversion to agricultural land takes place, soil conservation measures could counteract
the loss of soil quality (Veldkamp et al., 2020). But these measures are rare in the Eastern Congo
Basin due to poverty of subsistence farmers, socio-economic instability and a lack of
governmental intervention (Heri-Kazi Bisimwa and Bielders, 2020). Soil tillage and harvesting
further degrade the nutrient containing litter and topsoil layers. In consequence, fields often have
to be abandoned after only a few decades of use and recover only poorly (Carreño-Rocabado et
al., 2012; Ewel et al., 1991; Hattori et al., 2019; Heinrich et al., 2020; Kleinman et al., 1996;
Lawrence et al., 2010).
**1.2 Tropical soils responding to disturbance**
With the expansion of cropland into forested landscapes soil erosion rates are expected to
continue to increase. Soil erosion will undoubtedly impact biogeochemical cycles and change the
input, storage and exchange of C between soils and atmosphere as well as the flux of nutrients
between plants and soils in tropical systems in the region. To understand how tropical soils and
ecosystems respond to erosional disturbance, it is necessary to consider the combined effects of
climate, geology, topography, soil formation, biological processes and human disturbance. To
date, no study on the interrelationship of these controls on biogeochemical cycles has been
carried out in tropical ecosystems. However, studies carried out in other regions have shown that
controls on soil C dynamics, for example, are highly interlinked (Doetterl et al., 2015a; Hobley and
Wilson, 2016; Nadeu et al., 2015).
Soil redistribution as a consequence of erosion also changes the functionality of landscape units.
For example, soil degradation on hillslopes is matched by a rapid buildup of sediment deposits in
valley bottoms, where C and nutrient rich soil is rapidly buried in subsoils under new sediments.
While this consequence of deforestation can lead to an increase in the residence time of C due
to slower microbial C turnover in buried soil (Doetterl et al., 2012; Alcantara et al., 2017), important
nutrients are now lost to plants leading to biomass productivity (Veldkamp et al. 2020) and
degraded tropical forests generally negative for microbial processes in soils (Sahani & Behera,
2001). Soil redistribution is also known to change the temporal and spatial patterns of soil
weathering and affects C stabilization. In agricultural systems, the effects of this pressure can be





observed very clearly: erosion removes weathered soil from eroding slopes but also brings the
soil weathering front into closer contact with the C cycle (which occurs primarily in topsoils),
thereby affecting CNP cycling and the stabilization of C with minerals in these systems (e.g. Berhe
et al., 2012; Park et al., 2014; Doetterl et al., 2016).
Feedbacks on biogeochemical cycles between soil weathering, erosion will differ significantly not
only between natural and disturbed systems, but also between systems with differing soil mineral
reactivity. Recent advances have shown that mineral reactivity, constrained predominantly by soil
weathering and the mineralogy of the soil parent material, has direct control over soil organic
carbon, with climate exerting only indirect control through its impact on biogeochemical processes
and matter fluxes (Doetterl et al., 2015a; Tang and Riley, 2015). However, the exact effects of
mineralogy on the temperature sensitivity of microbial decomposer communities and the primary
productivity of ecosystems have, to date, not been constrained (Hahm et al., 2014; Tang and
Riley, 2015).

**1.3 Importance and outlook of research on the future of tropical biogeochemical cycles**

Tropical Africa is expected to experience great changes to both soil biogeochemical cycling and
ecosystem level carbon (C) fluxes between soil, plants and the atmosphere, with unknown
consequences for biogeochemical cycles. Despite decades of recognizing their importance,
tropical soils remain among the least studied in the world (Mohr and van Baren, 1954; Mohr et
al., 1972; Ssali et al., 1986; Juo and Franzluebbers, 2003). Although a more complete
understanding on soil-plant coupling in tropical environments is critical, most of our process
understanding on biogeochemical cycling between plant and soil is still derived from temperate
regions. However, due to differences in their environmental setting and soil forming history, many
tropical soil systems will likely react very differently to soil disturbance and land conversion than
temperate soil systems. For example, temperate ecosystems can differ fundamentally in the way
nutrients cycle and in the dominating and limiting factors for plant growth (Du et al., 2020). In
contrast to soils in the temperate zone, long lasting chemical weathering has led to a massive
depletion of mineral nutrients from soils in many tropical systems, although the remaining
available nutrients are very efficiently re-cycled in natural tropical biospheres (Walker and Syers,
1976; Vitousek, 1984). Hence, any loss of nutrients is therefore a critical disturbance with direct
effects on the functioning of tropical (agro-)ecosystems. Recent studies highlight the importance
of soil degradation and the change in chemical soil properties that follows land conversion on
plant communities in tropical systems (Bauters et al., 2021), organic matter turnover by microbial




decomposers (Kidinda et al., 2020 in review; Bukombe et al., 2021 in review) and the stabilization
of C and nutrients in soil of varying mineralogical properties (Reichenbach et al., 2021 in review).
Improving our process understanding on the consequences coupling between soil
biogeochemistry and plant responses in the context of tropical land use changes of land use
change on plant-soil interactions will help to better constrain plant-soil interactions in ecosystem
and land surface models and to better inform policy makers and stakeholders in improving land
management practices.

**1.4 Objectives and framework**

In the following we aim at providing an overview on the data collected by project TropSOC which
is now available to the research community as an open access database. We give a brief
description of the project's design before elaborating the structure of the database and its content.
Note that beyond the overview information presented here, more details to methods and sampling
designs for each assessed parameter is explained in great detail in the supplementary metadata
files accompanying the database.
The main objective of project TropSOC was to develop a mechanistic understanding of plant and
microbial process responses to changing soil properties in the African Tropics exemplified along
land use, erosional and soil geochemical gradients studied in the Congo and the Albertine Rift.
Trying to understand biogeochemical cycling affected by human activities in tropical (agro-
)ecosystems as a whole, TropSOC had two main foci:
(i) investigate how nutrient fluxes and organic matter allocation between tropical soils, plants differ
in relation to the controlling factors geochemistry, topography and land use.
(ii) investigate how the geochemistry of soils and their parent material control, interact with or
mediate the severity of erosional disturbance on C cycling in tropical soils.
In order to address these objectives, project TropSOC investigates effects on tropical soil
biogeochemical cycling and biological responses to variation in soil and environmental properties
along three main vectors (Figure 1): (i) Mineralogy of parent material, since it may drive the the
geochemical features of soils developed which control soil fertility and the potential of soils to
stabilize organic matter and nutrients. (ii) Landform, since topography may influence water and
soil fluxes, particularly erosional soil loss on slopes and soil deposition in valleys. (iii) Vegetation
and land cover, since it may control the input to and extraction of organic matter from soil, and



respond to variation in soil properties and hydrology, as well as mediate the impact of rainfall to
induce soil erosion.

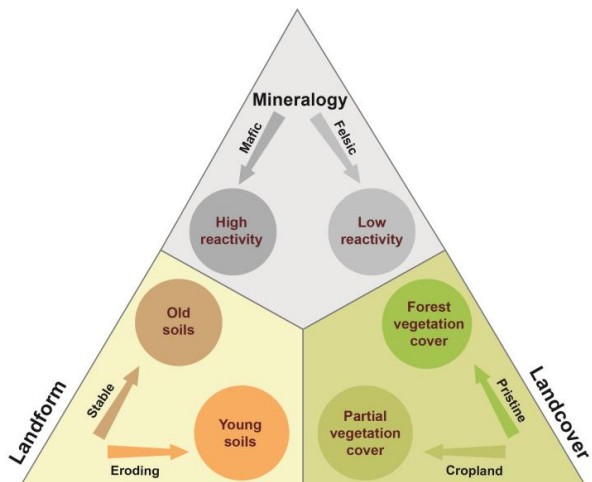

**Figure 1**. Factorial design of the project TropSOC studying biogeochemical cycles in Central African tropical forest and agricultural landscapes in relation to mineralogy, landform and land cover types.

Conducted in one of the hotspots of Global Change, the Central African Congo Basin and African
Great Lakes region the database described here is the foundation for several manuscripts
published as a part of the 2021 special issue "*Tropical biogeochemistry of soils in the Congo*
*Basin and the African Great Lakes region*" in SOIL Journal (Bukombe et al. 2021, in review;
Kidinda et al. 2020; Summerauer et al. 2021 in review; Reichenbach et al. 2021 in review; Wilken
et al. 2020 in review).
**2. Study and sampling design**
**2.1 Study area - Climate, topography, land use**
The study area of TropSOC is located in the eastern part of the Democratic Republic of the Congo,
Rwanda and Uganda, in the border region between the Congo and the Nile basin (Figure 2). It is
yet largely understudied (Schimel et al., 2015) despite its great significance for the global climate
system (Jobbágy and Jackson, 2000, Amundson et al., 2015) and being confronted with rapid





land conversion (Hansen et al., 2013) and forest degradation). The Climate of the study region is
classified as tropical humid with weak monsoonal dynamics (Köppen Af - Am) and mean annual
temperatures (MAT) ranging between 15.3 and 19.3 °C and mean annual precipitation (MAP)
between 1498 and 1924 mm (Fick & Hijmans, 2017) with high potential erosivity (Fenta et al.
2017) (Figure 2d).
As a part of the Eastern African Rift Mountain System, the active tectonism within the study region
produced a hilly, patchy landscape with steep slopes up to 60% and soil parent material ranging
from volcanic ashes to mafic and felsic magmatic rocks as well as a sedimentary rocks of varying
geochemistry and texture (Schlüter 2006) (Figure 2a,b).
The study area is dominated by agricultural land use, with larger patches of protected, old growth
closed canopy forest in highland areas (Figure 2c). Typical crops planted for subsistence farming
are rotations of cassava (*Manihot esculenta*), maize (*Zea mays*) and a variety of legumes and
vegetables. The dominant vegetation in all studied forests of the region is characterized as tropical
mountain forest (Verhegghen et al. 2012; van Breugel et al. 2015). Note that while forest
vegetation is thought to be largely spared from direct disturbance by human activities, large
mammal populations (i.e. African forest elephants, Great Apes) became extinct or largely reduced
due to hunting during the 20th century resulting in a massive increase in understory.


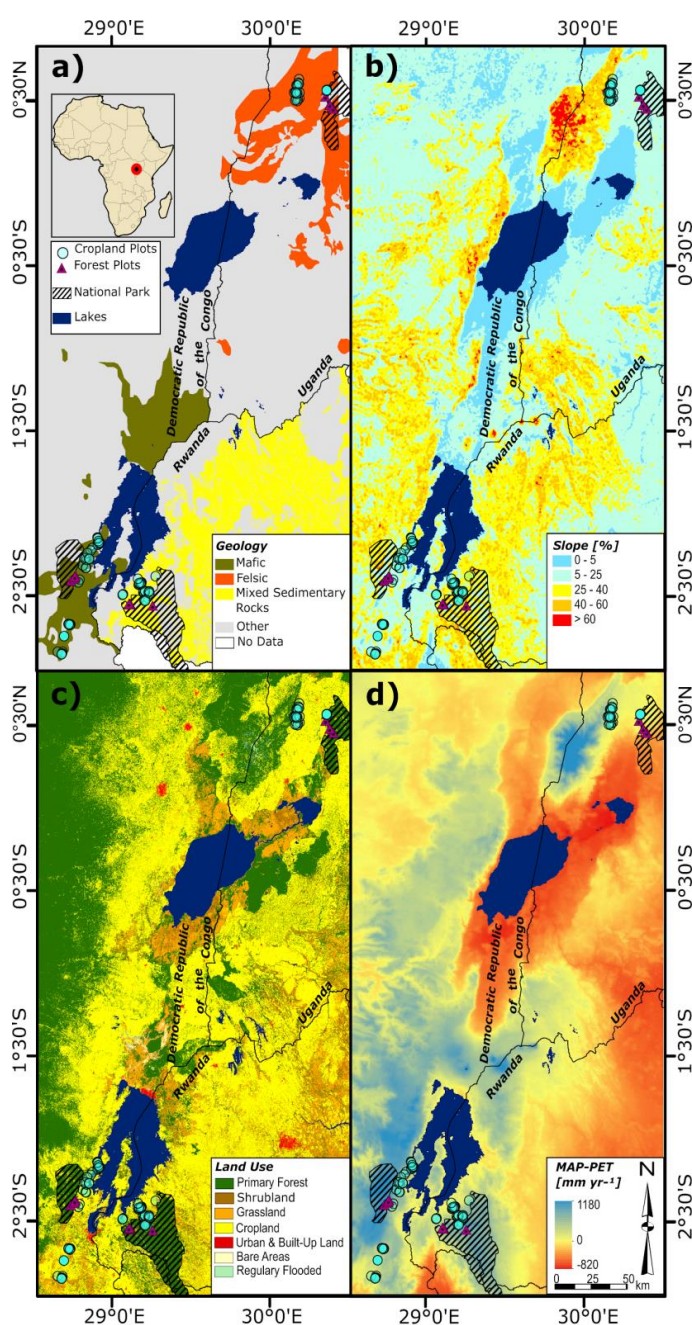

**Figure 2.** Overview of the study region with respect to major investigated factors: soil parent material geology and geochemical regions (a), slope steepness (b), land use (c) and climate d).





**2.2 Study area - Geochemistry and soil types**
Within the study area three regions each representing a geochemical differing parent material for
soil formation were determined. The first region (Figure 2a) is predominantly situated on mafic
magmatic rocks, typically mafic alkali-basalts ranging in age between 9-13 Ma (Schlüter 2006),
resulting from extinct (Mount Kahuzi) and active (Mount Nyiragongo) volcanic activities between
the cities of Bukavu and Goma, Kivu, DRC. The second region is situated on felsic magmatic and
metamorphic rocks typically consisting of gneissic granites ranging in age between 1600-2500
Ma (Schlüter 2006) near the city of Fort Portal on the foothill of the Rwenzori Mountain range,
Uganda. The third region is situated on a mixture of sedimentary rocks of varying geochemistry
consisting of alternate layers of quartz-rich sandstone, siltstone and dark clay schists ranging in
age between 1000-1600 Ma (Schlüter 2006) and spread across the Western Province of Rwanda
in and around the district of Rusizi.
The dominant soil types of the study region are various forms of deeply weathered tropical soils
(FAO, 2015). Potential ash deposition through the region's active volcanism occurs frequently,
re-fertilizing soils to various degrees. Following World Reference Base (WRB) soil classification
(IUSS WRB, 2015), soils in the mafic region can be described as umbric, vetic and geric Ferralsol
and ferralic vetic Nitisol. Soils in the mixed sedimentary rock region and the felsic region can be
described as geric and vetic Ferralsol. Soils in valley bottoms can locally show gleyic features,
where the dominating soil types are variations of fluvic Gleysol.
Several striking differences in the elemental composition of the three parent materials can be
noted. In the mafic region, bedrock is characterized by high iron (Fe) and aluminum (Al) content
as well as a comparably high content of rock-derived nutrients such as base cations and
phosphorus (P). The felsic and the sedimentary rock regions are characterized by lower contents
of Fe, Al as well as lower rock-derived nutrients contents and characterized by higher Si content
(Figure 3). A specific feature of the sedimentary site is the presence of fossil organic C in the
parent material of soils ranging between 1.29 - 4.03% C. Fossil organic C in these sediments is
further characterized by a high CN ratio (mean ± standard deviation: 153.9 ± 68.5), depleted in N
and free of $^{14}$C (due to the high age of sedimentary rock formation). The elemental composition
of soils at stable landscape position between the three regions retains the geochemical features
of its parent material to some degree and illustrates the process of enrichment of metal oxy-
hydroxides and the depletion of silica as a consequence of weathering. Generally, differences in
the elemental concentrations between the three regions are less pronounced in soil (figure 4)
compared to differences in parent material (figure 3). Remarkably, levels of rock-derived nutrients
in soil, while overall depleted compared to the parent material, are comparably similar, potentially
indicating biological mechanisms that keep these important nutrients in the plant-soil system
against a general trend of leaching and depletion, typical for weathered, old and nutrient poor
tropical soils (Grau et al., 2017 and references therein).


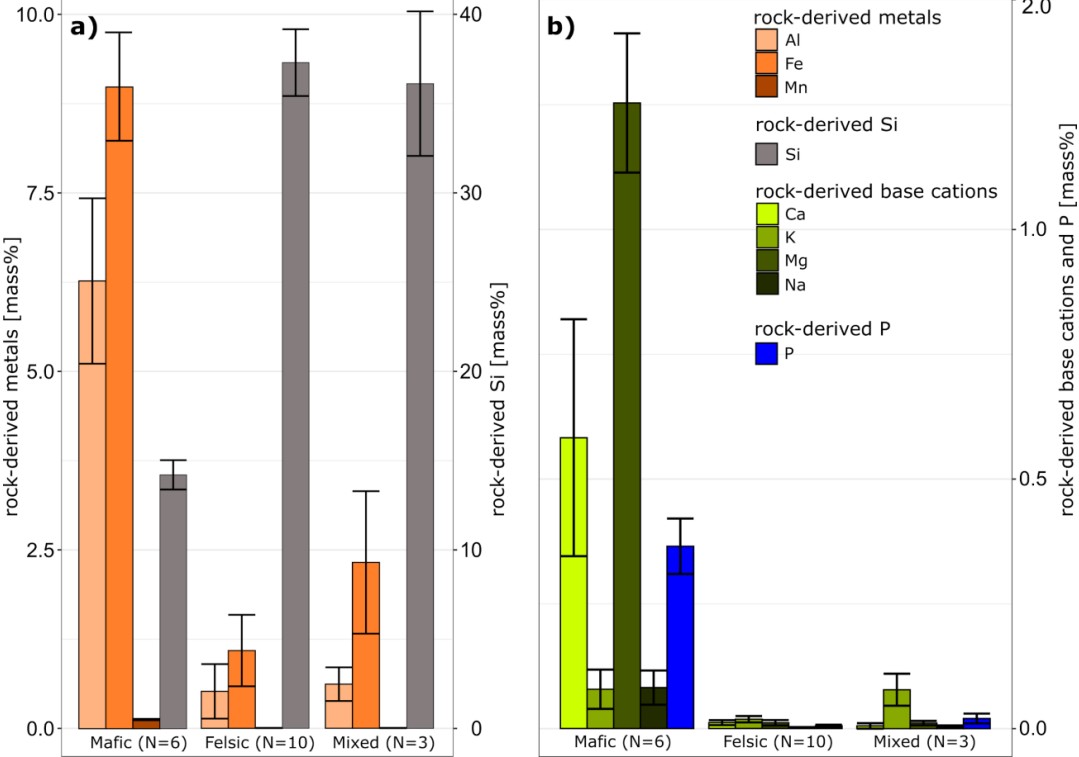

**Figure 3.** Chemical composition of unweathered rock samples representing the parent material for soil formation in three studied geochemical regions (mean +/- standard error). Panel 3a shows the distribution and concentration of rock derived aluminum (Al), iron (Fe) and manganese (Mn) and total silica content (Si). Panel 3b shows the distribution and concentration of rock derived calcium (Ca), potassium (K), magnesium (Mg), sodium (Na) and phosphorus (P). Note the difference in scale on y axis between panel 3a and 3b.



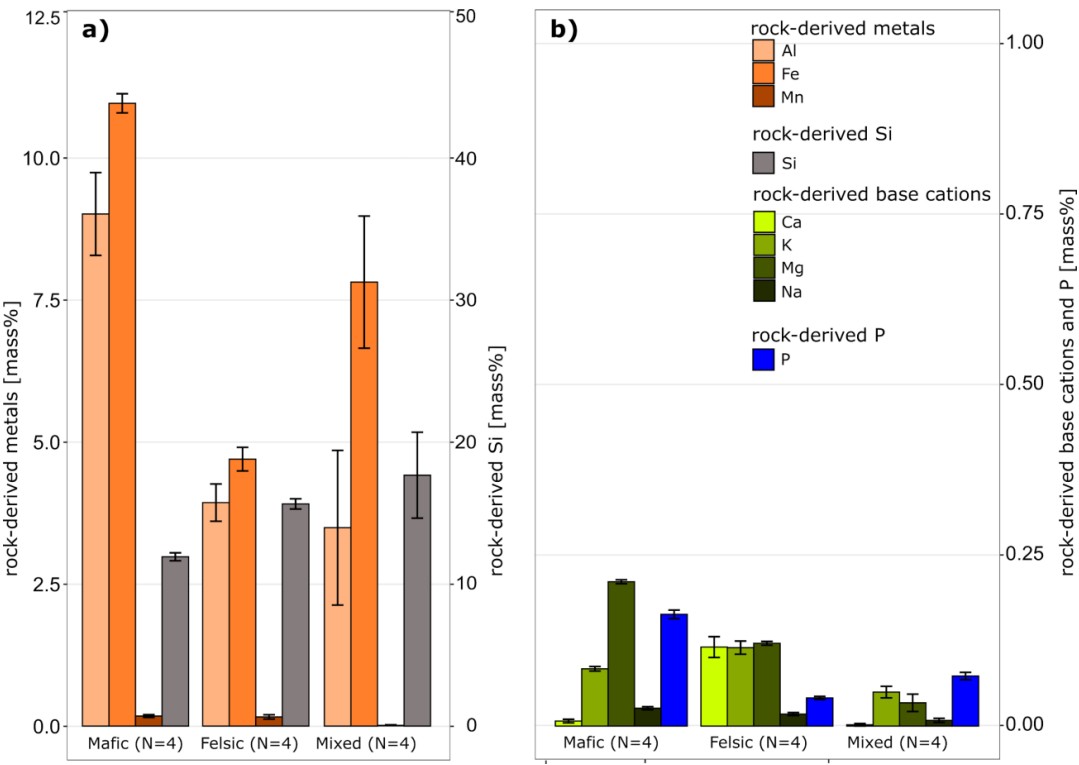

267

**Figure 4.** Soil chemical composition of subsoil in stable, old growth closed canopy forests (no erosion) in the three investigated geochemical regions (mean +/- standard error). The data illustrates the convergence of elemental concentrations between the three regions as a result of weathering and soil development. Abbreviations explained in figure 3. Note the difference in scale on y axis between panel 4a and 4b.






In summary, the study region provides a unique combination of (i) near-pristine forest and
agricultural land use, (ii) steep terrain and heavy tropical precipitation with high erosion potential
and (iii) geologically diverse parent material for soil formation. These factors make the study
region ideal for identifying the importance of various controls on tropical soil biogeochemical
cycles.

**2.3 Overview to plots and sampling design**
Plots were established along geomorphic gradients in old-growth closed canopy forest as well as
cropland in all three geochemical regions. Field campaigns to collect soil and plant samples at
136 forest and cropland plots along slope gradients (catena and stratified random approaches)
and additionally within several cropped nearby micro-catchments were carried out between March
2018 and July 2020. A detailed description on data quantity and quality can be found in the
metadata files accompanying the database and are briefly described in section 4.1 of this
publication. In order to cover potentially stable, eroding and depositional landforms, topographic
positions of plots ranged from plateaus (slope < 5%), over two slope positions (slopes between 9
and 60%) to valley positions (slopes < 5%) (Table 1).

**Table 1.** Topographic information of TropSOC plots across different geochemical regions and land use. Slope and altitude are displayed as minimum and maximum values. Each topographic position per geochemical region contains the range between 3-7 field replicate plots.

| | **felsic region (Uganda)** | | | | | |
|---|---|---|---|---|---|---|
| | **forest plots** | | | **cropland plots** | | |
| *topographic position* | *plateau* | *sloping* | *valley* | *plateau* | *sloping* | *valley* |
| slope [%] | 3 - 5 | 9 - 55 | 3 | 1 - 5 | 7 - 50 | 1 - 5 |
| altitude [m] a.s.l | 1304 - 1306 | 1271 - 1420 | 1272-1277 | 1507 - 1797 | 1466 - 1830 | 1587 - 1768 |
| | **mafic region (DR Congo)** | | | | | |
| | **forest plots** | | | **cropland plots** | | |
| *topographic position* | *plateau* | *sloping* | *valley* | *plateau* | *sloping* | *valley* |
| slope [%] | 3 | 11 - 60 | 1 - 2 | 0 - 5 | 8 - 43 | 0 - 3 |
| altitude [m] a.s.l | 2208 - 2227 | 2188 - 2248 | 2181 - 2310 | 1477 - 1731 | 1486 - 1774 | 1505 - 1708 |



| topographic position | mixed sedimentary region (Rwanda) | | | | | |
|---|---|---|---|---|---|---|
| | forest plots | | | cropland plots | | |
| | *plateau* | *sloping* | *valley* | *plateau* | *sloping* | *valley* |
| slope [%] | 3 | 9 - 60 | 1 | 3 - 5 | 8 - 50 | 2 - 5 |
| altitude [m] a.s.l | 1908 - 1939 | 1891 - 2395 | 1882 - 1889 | 1719 - 1837 | 1565 - 1952 | 1556 - 1758 |

**2.4 Sampling design forest**
*2.4.1 Forest plot installation*
Sampling in forests followed a strict catena approach and plots were established following an
international, standardized protocol for tropical regions (Phillips et al. 2016). Within each
geochemical region, three plots covered by old-growth closed canopy tropical forest vegetation
(forest that developed a complex structure characterized by large, live and dead trees) were
established per topographic position as field replicates representing an area of 40 m x 40 m per
plot were established from February to June 2018. Each plot was subdivided in four 20 m x 20 m
subplots and a total of 36 forest plots were established this way (four topographic positions with
three replicate plots each in three geochemical regions). Note that three plots in the mafic region
had to be relocated due to safety reasons after the sampling period. For an overview on forest
plot sampling design see Figure 5a.
*2.4.2 Sampling mineral and organic soil layers*
At the time of plot installation, four replicate soil cores per plot (one in each subplot) were taken
in a depth-explicit way in 10 cm increments up to 1 m soil depth, and combined as composites
per plot. In addition, one soil profile pit was dug to a depth of 100 cm in the center of one of three
replicate plots (Figure 5) per topographic position in each geochemical region. These soil pits
were dug and described according to FAO guidelines (FAO, 2006).
Leaf litter (L horizon) and partially decomposed organic material in O horizons were sampled at
eight points along the border and in the center of each forest plot (Figure 5a) at the time of soil
sampling. At each sampling point, the thickness of the L and O horizon layer were measured with
a ruler and then sampled within a 5 cm x 5 cm square. When the litter layer was too thin (= no
closed coverage of forest floor with litter), the sampling square was expanded to a 10 cm x 10 cm



to retrieve enough sample material. The nine samples of each layer per plot were combined to
one composite sample.
All collected composite samples were kept cooled until being brought to the laboratory (usually
within 48 hours). In the laboratory, samples were oven-dried at 40°C for 48-96 hours and then
weighed (accuracy: +/- 0.01 g). Derived soil parameters are detailed in section 2.7.
### 2.4.3 Forest inventory and aboveground standing biomass
In 2018, full inventories of the forest tree species and standing aboveground biomass (AGB) were
conducted on all forest plots. The forest inventory followed an international, standardized protocol
for tropical regions (Matthews et al., 2012). First, we identified the species of all living trees with
a diameter at breast height (DBH, measured at 1.3 m above ground) greater than 10 cm in each
plot. Second, these identified trees were classified into the following empirical DBH classes: 10 –
20 cm, 20 – 30 cm, 30 – 50 cm and > 50 cm. Third, to estimate the above-ground biomass (AGB),
we constructed stand-specific height diameter (H–D) allometric relationships using a
representative subset of the plot-specific trees (Méchain et al., 2017). For this, 20% of all
measured, specific trees were selected for height measurement, across the DBH range that was
recorded per plot. Depending on the tree abundance of each DBH class, the height of three to
five individual trees were then measured using a hypsometer (Nikon Laser Rangefinder Forestry
Pro II, Nikon, Japan). AGB for each individual tree was then estimated using the allometric
equation as described by Chave et al. (2014) for moist tropical forests. To estimate wood density
data, we used species averages from the DRYAD global wood density database (Zanne et al.,
2009). To extrapolate this information for the entire plot for all our sites, we applied a stand-
specific height-diameter regression model; modelHD, available within the R package BIOMASS
(Méchain et al., 2017). In a last step, aboveground standing biomass carbon stock was estimated
assuming that that all samples standing biomass has a 50 wt.% share of C (Chave et al., 2005).
A re-census was carried out in 2020, in order to detect changes in above-ground standing
biomass and to determine tree mortality. Tree mortality rate ($\lambda$) at each plot was assessed
following Lewis et al. (2004), using inventories conducted in 2018 and 2020. Tree mortality rate
was calculated for all tree stems with DBH>10cm in every plot.
### 2.4.4 Canopy leaves
To assess plant functional traits (leaf nitrogen,  phosphorus, potassium, magnesium and calcium
content)  of living canopy leaves (see section 2.7), we sampled, at the beginning of the weak dry





season (December-February), sun-exposed shoots from the outer canopy of selected tree
species that collectively make up 80% of the standing basal area per plot with the help of trained
tree climbers and following a sampling protocol described in Pérez-Harguindeguy et al. (2016).
For every tree species, we selected at least 3 individual trees, and a minimum of five and
maximum of 17 trees per plot were sampled for mature, healthy-looking (= without signs of
herbivory) individual canopy leaves. Where sampling of outer canopy leaves was physically not
feasible, partially shaded leaves situated below the uppermost canopy were sampled.
**2.5. Sampling design cropland**
***2.5.1 Cropland plot installation***
Plots on cropland were established following a stratified random approach using the same slope
classification and selection criteria as for forest sites. However, cropland plots belonging to the
same geochemical region and topographic position were not connected along a hillslope catena.
On cropland only fields that were currently covered by cassava were sampled. Cassava fields
were chosen since cassava is one of the most important food crops in the region, harvested for
both tubers and leaves. Rotations of cassava, maize, pulses and vegetables are common
throughout the area and two harvests are possible per year. The main varieties of cassava on our
sites were Mwabailon, Nabiombo, Mwamizinzi, Sawasawa (in Eastern DRC), Bukalasa,
Shayidire, Gitamisi, Amaduda (in Rwanda), Sambati, and Mubalaya (in Uganda). Only fields
without soil protection measurements (i.e. terraced systems) were sampled. For an overview on
forest plot sampling design see Figure 5b.
***2.5.2 Soil sampling***
Soil sampling was carried out in the same way as for forest soils with the exception that only two
cores were combined per plot taken within a 3 m x 3 m area to create depth explicit composite
samples. A total of 100 cropland plots were sampled this way (Figure 5) with 3-7 field replicate
plots per topographic position (plateaus, slopes, valleys) in each geochemical region. No L and
O horizons were present in cropland, and no soil profile description was carried out. Derived soil
parameters are detailed in section 2.7.
***2.5.3 Biomass and crop yield***
As part of the regional stratified random sampling design for cropland plots (see cropland plot
installation), biomass from different cassava varieties was collected for 65 plots out of the 100



sampled cropland plots. Biomass was sampled shortly before harvest, approximately at the time
of the plant tuber's maximum development. The timing of harvest differed between 12 - 24 months
after planting depending on the variety and season. Within each plot, a 3 m x 3 m sampling area
was chosen close to the center of each field and all cassava plants in this area were counted and
harvested. The biomass of all plants was separated into leaves, stems and tubers. These parts
were then weighed separately and individually at the time of sampling (i.e. in a field moist state).

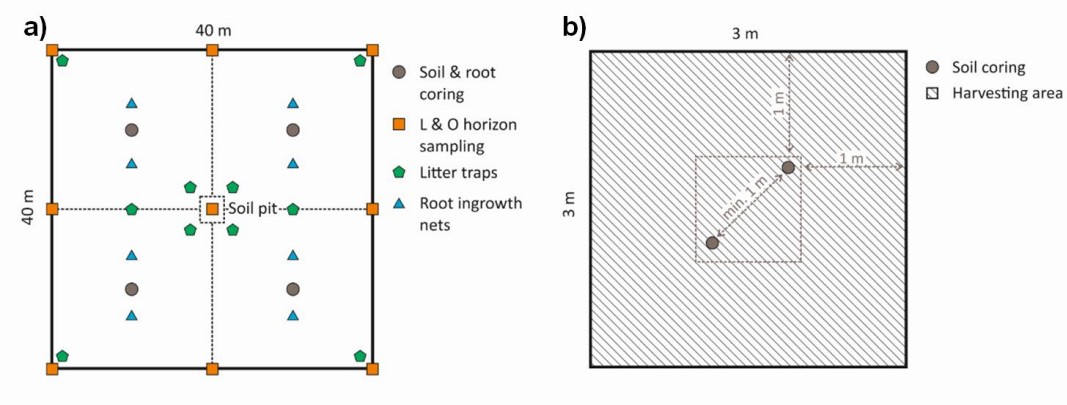


**Figure 5.** Overview on forest (a) and cropland (b) plot sampling design. Forest plots were
subdivided into four 20 m x 20 m subplots and one soil profile pit was established per topographic
position in each geochemical region for one of three replicate plots.
***2.5.4 Land use history and management assessment***
Farmers were sent a questionnaire to collect information on the land use and management history
of sampled fields following McCarthy et al. (2018). This questionnaire was completed for a
corresponding total count of 87 out of the 100 sampled cropland plots.
**2.6 Monitoring design**
***2.6.1 Micrometeorological data***
Three weather stations (ATMOS 41, Meter, Germany) were installed in August 2018 in each
geochemical region of project TropSOC close to the investigated forest catenae (mafic: latitude:
-2.324457° / longitude: 28.740818°; felsic: latitude: 0.561767° / longitude: 30.356808°, mixed
sedimentary rocks: latitude: -2.460503° / longitude: 29.095251°). An additional weather station
was installed in the mafic region near a cropland catchment, (latitude: -2.583984° / longitude:
28.715298°) which was selected for high-resolution erosion monitoring (see Wilken et al. 2021).



Furthermore, a meteorological station in the city of Bukavu (latitude: -2.499979°, longitude:
28.845009°) and Lukananda (latitude: -2.344073°, longitude: 28.750937°) were put into
operation. All stations collected data at a temporal resolution of 5 minutes on precipitation, air
temperature, relative humidity and air pressure. Additionally, global radiation and wind speed
were measured at stations Bukavu and Lukananda.
### *2.6.2 Litterfall sampling*
Litterfall was assessed following a standardized protocol to measure tropical forest carbon
allocation and cycling (Matthews et al., 2012). At each of our 36 forest soil sampled plots, 10 litter
traps were installed and distributed evenly and systematically per plot. These had a diameter of
60 cm each and were installed at a height of 1.0 m above ground. Litter samples were collected
every two weeks for the period between August 2018 and February 2020 and later aggregated,
to assess seasonal and annual variability in litter productivity and quality (see section 2.4).
Collected litter included all organic residues collected by the traps. Larger, dead animals and
woody material > 2 cm in diameter were discarded. After sampling, material from all 10 traps per
plot was mixed to obtain a composite sample. These composite samples were taken to the
laboratory the day of sampling, oven-dried at 70°C for 72 hours and subsequently weighed (dry
weight, accuracy: +/- 0.01g). Data is provided as Mg ha$^{-1}$ day$^{-1}$ per plot and as the sum of total
litter production per plot, aggregated at the seasonal level and annual level. The considered
seasons were categorized based on the average precipitation for each period: weak dry season
(December-February), strong rain season (March-May), strong dry season (June-August) and
weak rain season (September-November).

### *2.6.3 Belowground standing root biomass*
For all soil sampled forest plots, standing root biomass and fine root production were assessed
from September 2018 to December 2019. Sampling took place once per season within this period
(one coring every three months) and a total of three rain seasons and three dry seasons) in 2018
and 2019 were covered. Each plot was divided into four equally sized subplots of 20 m x 20 m.
Prior to deciding the root sampling strategy and size of depth intervals, root distribution was
assessed using soil profiles that were dug in the plot centers for soil classification purposes. This
assessment revealed that roots mostly dominated the organic horizons and the upper 50 cm of
mineral soil (data not shown).




Belowground standing root biomass was sampled using a soil core sampler (Vienna Scientific
Instruments, Austria). Two cores were sampled per subplot where undisturbed soil cores were
divided into five depth layers: one organic soil layer (O horizon), and four mineral soil layers from
0 – 10 cm, 10 – 20 cm, 20 – 30 cm, 30 – 50 cm. After transport to the laboratory, each sample
was rinsed inside a 2 mm sieve; roots were separated into fine roots (<= 2 mm diameter) coarse
(> 2 mm diameter) using calipers. In addition, fine and coarse roots were separated into living and
dead roots based on criteria such as color, root elasticity and the degree of cohesion of cortex,
periderm and stele; i.a. roots were considered living when root steles were bright and resilient
(Ostonen et al., 2005). The dry mass of isolated roots per plot was assessed after previously
having dried the root samples at 70 °C for 72 hours. Data is provided as mg cm$^{-3}$ per plot per
sampling date and is also aggregated at the seasonal and annual level.

***2.6.4 Fine root net primary production***
Fine root net primary productivity was assessed using the ingrowth net method following (Ohashi
et al., 2016). Two net sheets (polyester mesh aperture size 2 mm, 10 cm wide, 20 cm high) were
installed per subplot in a regular pattern with a distance of approximately 1 m between the two
nets. Each net was vertically inserted in the top 20 cm of soil starting from the surface of the
mineral layer. Nets were sampled every three months after installation and seasonally four times
a year, from September 2018 to December 2019. Data is provided as g m$^{-2}$ and g m$^{-2}$ day$^{-1}$ of
total fine root production per plot over a certain period of time, and also provided aggregated at
the seasonal and annual level.

**2.7 Chemical and physical analyses**
A wide range of chemical and physical parameters were assessed for the sampled soil and plant
material with the aim to (i) characterize indicators of soil redistribution, (ii) the degree of soil
weathering, (iii) the physical structure of soil as well as (iv) soil fertility and (v) soil organic carbon
characteristics in order to link them to (vi) functional traits of the sampled biomass, (vii) biomass
production and (viii) land management. For a full overview of all assessed parameters including
their assessment methods, please consult the metadata accompanying the database.



Among others, key measured parameters encompass:

***Basic physical parameters***
-      Soil bulk density
-      Soil texture
-      Soil water holding capacity
***Basic chemical parameters***
-      Soil pH (KCl)
-      Soil potential cation exchange capacity and its base saturation
-      Soil effective cation exchange capacity and its base saturation
-      Main elemental composition of bulk soil (Al, Fe, Mn, Si, Ti, Zr, P) and the total reserve in
base cations (Ca, Mg, Na, K) in rock parent material, soil, litter and vegetation samples

-      Pedogenic oxides concentration (Al, Fe, Mn)
***Available nutrients***
-      Dissolvable soil organic nitrogen and carbon
-      Plant available phosphorus in soil
***Organic matter characteristics***
-      Total and organic carbon and nitrogen content in rock parent material, soil, litter and
vegetation samples

-      Bulk soil radiocarbon signature
-      CN ratio in soil, litter and vegetation samples
-      Soil carbon stabilization mechanisms



***Microbial activity***
-      Heterotrophic soil respiration (including isotopic signature of respired gas)
-      Microbial biomass during incubation
-      Extracellular enzyme activity during incubation
***Soil redistribution***
-      239+240 Pu activity
All of the parameters listed above have been measured in soil for three depth layers (0-10 cm,
30-40 cm, 60-70 cm) representing distinct sections of the soil profile. Physico-chemical key
properties of the remainder of soil samples in other soil layers have been assessed using mid-
infrared spectroscopy and predicted following the workflow of Summerauer et al., 2021 in review).
An overview of chemical and physical key soil parameters is provided in Appendix Table A1. Note
that all physico-chemical soil properties and the corresponding mid-infrared data are part of the
central African spectral library (Summerauer et al., 2021 in review) and minimize the need for
future traditional soil analyses.

**2.8 Milestones reached**
Overall a total of approximately 2100 soil and rock samples were collected, of which about 10 -
30% were used yet for detailed analyses in different experiments by our group (see below).
Additionally, 6000 above- and belowground biomass and litter samples were taken during several
sampling and monitoring campaigns at forest and cropland sites. Several thousand and mid-
infrared (NIR-MIR) spectra in the wavenumber range 600 cm$^{-1}$ to 7500$^{-1}$ (wavelength 1333.7 nm
- 16666.7 nm) were collected across the sampled plant and soil samples and were used to train
calibration models for each property to predict spatially and depth explicit soil parameters in
relation to soil fertility, carbon stocks and carbon stabilization using partial least square
regressions following the workflow of Summerauer et al., (2021 in review). Furthermore, since
2018, continuous monitoring has been carried out for the installed weather stations and vegetation
dynamics in tropical forests have been assessed from August 2018 until December 2019. Water



and heat fluxes between soil and atmosphere are monitored using several weather stations and
soil probes to monitor heat and water transfer into soil.
Analyses conducted on collected samples, so far, contributed to scientific advances realized
through
-  the creation of a data frame of reference samples for calibration used in the newly
developed soil spectral library for central Africa (Summerauer et al., 2021 in review).
-  an investigation on the role of geochemistry and geomorphic position for soil organic
matter stabilization mechanism and patterns of SOC stocks in tropical rainforests
(Reichenbach et al., 2021 in review).
-  an investigation of the role of geochemistry and geomorphic position on the heterotrophic
soil respiration (Bukombe et al., 2021 in review) as well as the role of adaptations of
microbial communities and their strategies to access nutrients along the investigated
forest gradients (Kidinda et al., 2020 in review).
-  an assessment of the suitability and the application of radioisotope $^{239+240}$Pu inventories
for studying soil erosion processes in tropical forests and cropland (Wilken et al., 2020 in
review)
-  soil fractionation and incubation experiments encompassing cropland soils along
geomorphic and geochemical gradients (unpublished).
-  as part of this manuscript, the entirety of TropSOC's data is available as an open-access
database with extensive metadata documenting experimental approaches, framing of the
analyses, data quality and methodology. An overview of all datasets presented in this
database is given in Appendix Table A2.
In summary, TropSOC's first results demonstrate that even in deeply weathered tropical soils,
parent material has a long-lasting effect on soil chemistry that can influence and control microbial
activity, the size of subsoil C stocks, and the turnover of C in soil. Soil parent material and the
resulting soil chemistry need to be taken into account in understanding and predicting C
stabilization and turnover in tropical forest soils. Given the investigated rates of erosion on
cropland, our findings confirm the threat of large losses or organic matter leading to sharp decline
in soil fertility with little potential of soils to recover from nutrient losses naturally on decadal or



centennial timescales. TropSOC highlights that considering feedbacks between geochemistry
and topography to understand the development of soil fertility in the Afrcan Great Lakes Region
regions can significantly improve our insights into the role of tropical soils for reaching several key
sustainable development goals such as climate mitigation and zero hunger and help to raise
awareness for the need to maintain limited soil resources for future generations. Future work
realized in project TropSOC based on the database will provide further insights into biomass and
plant trait responses to soil geochemistry in forests, as well as cassava yield responses and SOC
dynamics in cropland along the investigated geomorphic and geochemical gradients across the
region.
**3. Structure of TropSOC project database (TropSOC v1.0)**
**3.1 Database hierarchy**
Datasets are given as tab-delimited .csv files. For each .csv file the metadata describing data
structure and assessment methods are given in a .pdf file of the same name. Moreover, additional
.pdf files for each main section of the database (basic information, forest, cropland, and
microscale meteorology) are given, providing an overview of the structure within each section.
Note that the '**basic information**' section of the database provides the linkages between
individual data, e.g. from soil analysis and the location and/or soil depths where these samples
were acquired (for linkages see also Figure 6).






**Figure 6.** Overview of linkages between datasets in the TropSOC database v1.0. Note that for
each data .csv-file an .pdf-file is given detailing the metadata of the respective data sheet.








**3.2 Database infrastructure**

***3.2.1 Basic information***

The database comprises basic information of all plots and single point sampling positions where data were collected during project TropSOC. An overview of the structure of the database is presented in Appendix Table A2. The basic information of the database is structured in the following way:

**Part 1** – Location and basic background information for all plots and points where data were collected. Data can be found in file *11_plots_points.csv,* with description given in *11_plots_points.pdf*.

**Part 2** – Sample identifier for the database' internal connection between location of plots, points and soil data from different soils depths as well as vegetation data. Data is stored in *12_sample_identifier.csv*, with description given in *12_sample_identifier.pdf*.

The key element to link all datatables for which data was collected and samples analyzed is the plot ID and its derivative the sample ID. This identifier allows to link the results from sample analysis with the locations given in *11_plots_points.csv*. This results in a n:1 connection between *12_sample_identifier.csv* and *11_plots_points.csv*. See metadata file 11_plots_points.pdf for an overview on the structure of the plots ID and 12_sample_identifier.pdf for an overview on the structure of the sample ID.

***3.2.2 Forest***

TropSOC's forest data consists of seven parts (Table A2 for overview) structured as paired .csv / .pdf files, containing the data (.csv) and accompanying metadata (.pdf) describing parameters and methods. Additionally, an overview to all collected forest data is given in file *2_forest.pdf.*

**Part 1** – Above and belowground vegetation data acquired in 2018, 2019 and 2020 at all forest plots, comprising 13 data sets (Dataset files 2.1.1 - 2.1.13).

**Part 2** – Mineral soil layer data acquired in 2018 at all forest plots, comprising 3 data sets (Dataset files 2.2.1 - 2.2.3).

**Part 3** – Organic soil layer data acquired in 2018 at all forest plots, comprising 1 data set (Dataset file 2.3).





**Part 4** – $^{239+240}$Pu soil inventory carried out in 2018. In contrast to part 1 to 3 of the forest data, Pu
data represents individual points and does not follow the plot concept in a strict manner (Dataset
file 2.4).
**Part 5** – Soil experiments carried out from 2018 to 2020, comprising 3 data sets with results from
laboratory soil incubation and fractionation experiments and additional data from soil sample
analyses (Dataset files 2.5.1 - 2.5.3).
**Part 6** – Parent material elemental composition analysed based on unweathered rock samples
taken within plots or from nearby road cuts and mines surrounding the study sites (Dataset file
2.6).

**Part 7** – Soil profile descriptions done in soil pits at the centre of plots following WRB-FAO soil
description (Dataset file 2.7).

***3.2.3 Cropland***
TropSOC's cropland data consists of the following seven parts (Table A2 for overview) structured
as paired .csv / .pdf files, containing the data (.csv) and accompanying metadata (.pdf) describing
parameters and methods. Additionally, an overview to all collected cropland data is given in file
*3_cropland.pdf*.
**Part 1** – Biomass and management data acquired in 65 and 87 out of 100 sampled cropland plots
respectively, comprising 2 datasets (Dataset files 3.1.1 - 3.1.2).
**Part 2** – Data on mineral soil layers was acquired in 2018 for 100 cropland plots and comprising
3 datasets (Dataset files 3.2.1 - 3.2.3).
**Part 3** – Pu soil inventory carried out in 2018. In contrast to part 1 and 2 of the cropland data, Pu
data represents individual points and not plots and was sampled across several catchments
(Dataset file 3.3).
**Part 4** – Soil experiments. This part of the database comprises 2 datasets with results from
laboratory soil incubation and fractionation experiments and additional data from soil sample
analyses (Dataset files 3.4.1 - 3.4.2).





### *3.2.4 Meteorological data*

The meteorological data comprises 4 parts (Table A2 for overview) structured as paired .csv / .pdf files containing the data (.csv) and accompanying metadata (.pdf) describing parameters and methods:

**Part 1**: Locations of meteorological stations: Coordinates, elevations and contact addresses for the respective data (Dataset file 4.1)*.*

**Part 2**: Daily meteorological data: six meteorological stations recording precipitation, air temperature, relative humidity, air pressure, solar radiation, wind speed (Dataset file 4.2).

**Part 3**: High resolution five-minute triggered precipitation data: Precipitation recorded at the time of tipping bucket tilt at a resolution of five-minutes resolution (Dataset file 4.3).

## 4. Database status

### 4.1 TropSOC v1.0

The current version, v1.0, of TropSOC includes several thousand individual plant and soil samples collected across 136 sites spanning cropland and forests in the East African Rift Valley System and a large variety of parameters. A total of 36 .csv datasheets is available that gives all analyses done for specific samples. Datasheets are structured according to the descriptions given in section 3 and described and elaborated on in the accompanying metadata files. The current distribution of data points across the various levels of the database hierarchy is shown in Table 2. All individual data entries present in the database have passed quality control done by experts that were involved in the creation of the data. Where applicable, reports on the quality assessment of each parameter can be found in the metadata .pdf files accompanying the .csv files.

**Table 2.** Overview on the current number of data points in TropSOC v1.0 on plant, soil and meteorological and their affiliation to the hierarchical levels forest and cropland. Numbers in tables refer to the number of data entries at the lowest available aggregation level (= highest resolution of data). For details on parameters, see the according metadata descriptions. Note that in the felsic (Uganda) and mixed sediment region (Rwanda) collected weather station data represents both cropland and forest while separate stations were available for the two land cover classes in the mafic region (DRC). Abbreviations: SOM = Soil organic matter.





| Plant-Soil observations | Plots | Bulk soil samples (0-100 cm soil depth, 10cm increments) | Bulk Vegetation samples (above/ belowground) | Incubated soil layers | SOM fractionated soil layers | Plots with vegetation assessments |
|---|---|---|---|---|---|---|
| Forest | 36 | 916 | 1437/4374 | 112 | 145 | 40 |
| Cropland | 100 | 1190 | 132/66 | 131 | 159 | 65 |
| **Total** | **136** | **2106** | **1569/4400** | **243** | **304** | **105** |
| **Meteorological observations** | **Stations** | **Precipitation** | **Air temperature** | **Relative humidity** | **Global Radiation** | **Wind speed** |
| Felsic region | 1 | 541 | 541 | 541 | 0 | 0 |
| Mafic region (forest) | 1 | 674 | 858 | 860 | 860 | 644 |
| Mafic region (cropland) | 3 | 1310 | 1310 | 1312 | 709 | 650 |
| Mixed sediment region | 1 | 90 | 520 | 565 | 0 | 0 |
| **Total** | **6** | **2615** | **3229** | **3278** | **1569** | **1294** |


**4.2 Accessing TropSOC v1.0 and reporting issues/ask questions to its hosting platform CBO**


Users may access the TropSOC database v1.0 and its supporting information through the
supplementary material provided as part of this submission. Version v1.0 of the database is also
available through the data download section of the Congo Biogeochemistry Observatory (CBO)
(https://www.congo-biogeochem.com/data) and the PANGEA open access environmental data
repository. CBO is a consortium of researchers who study biogeochemical cycles and
atmosphere-plant-soil interactions in tropical Africa with a focus on the Congo Basin and the
African Great Lakes region (Doetterl et al. 2020). Within CBO's framework, a multinational group
of young scientists from Africa, Europe and the United States conducts cross-disciplinary
environmental research across tropical Africa but with focus on the Congo basin. The dedication
of young African researchers to understand and preserve the threatened natural resources of
their home countries is paired with the resources of some of the most experienced and largest
research groups focusing on African tropical forest and agroecosystems. Founded in 2018 by
scientists of several African and European institutions and support by multinational organization
such as CGIAR-IITA and CGIAR-ICRAF, CBO has become an important scientific network in
tropical Africa for studying biogeochemistry in soils and sediments creating synergies between
local key institutions and international researchers, crucial for the implementation of research in
remote and difficult to access environments. Research at CBO is funded and supported by
German, Belgian, US and Swiss Research foundations and linked to research institutes at Ghent
University, Augsburg University, Florida State University, ETH Zurich, the University of Louvain
and the Max Planck Society.



Users are encouraged to provide feedback and corrections to existing data if problems are
discovered by contacting CBO (contact@congo-biogeochem.com) or the corresponding author
of this manuscript (sdoetterl@usys.ethz.ch). Corrections will be implemented in consecutive
versions of the database that can be downloaded via the CBO site.
**4.3 Consecutive database versioning and archiving**
Updated versions of the database will be periodically released following either substantial
changes or new peer-reviewed publications, leveraging the dataset. Versioning of these official
releases are tracked using an associated version number, e.g. TropSOC v1.0, and so on. These
official releases will be archived at ETH Zurich's Research collection via ETH's Soil Resources
Group    (https://soilres.ethz.ch/)    and    the    CBO    data    storage    (https://www.congo-
biogeochem.com/data) with a dataset DOI issued for each release via ETH Zurich so that users
may revert back to the earlier version if so required. These archived releases will be maintained
into perpetuity to facilitate reproduction of any analyses conducted using a past version of the
database. When accessing the dataset and using it for own research, users commit to cite the
original manuscript provided here in addition to the version number, DOI and any description
provided to future versions of the database (see section 6 for details).
**5. Database governance and participation**
TropSOC is a community effort with multiple contributors operating at different levels (Figure 7).
Governance of TropSOC is required in order to ensure continuity of services and to plan for the
future evolution of this data repository. Studying the rapid environmental changes to the African
Tropics is a central research objective for the scientists of the Congo Biogeochemistry
Observatory (CBO) making it the ideal body to govern future versions of TropSOC. The
governance structure of TropSOC is briefly described in Figure 7. While the TropSOC core team
is responsible for the original version of the database, its maintenance, management and
archiving, scientists involved in the Congo Biogeochemistry Observatory (CBO) oversee the
establishment of cooperative agreements on the long term and act as a steering committee for
modifications on TropSOC suggested by the research community. The main role of the steering
committee is to determine the feasibility of major changes to TropSOC proposed by the
community and to coordinate activities that would like to build upon TropSOC or continue similar
research work within the framework of CBO. Although the structure of TropSOC is oriented
around individual and research projects, the nature of scientific research is often more group-





focused. For example, teams of researchers generally work together to seek out funding and to
conduct research. Thus, in some cases a group or team of individuals may seek to utilize or
modify TropSOC for their purposes. Such groups can petition the scientific steering committee to
be formally designated a CBO member group. Approved organizations should nominate a
member to serve on the steering committee.
Interested researchers are also invited to contribute data to future versions of TropSOC in order
to grow the database. Anyone can be a data contributor provided they agree to the terms of use
and follow the proper steps for contributing data to TropSOC. If such suggestions arise, the CBO
steering committee together with the TropSOC core team are responsible for approving the
suggested changes and additions to the database. Upon approval, the TropSOC core team will
interact with the new data contributors to implement the suggested data additions. In the case of
organizations or individuals making larger changes or additions to TropSOC, a designated data
maintainer from new contributor groups is required to coordinate the technical aspects of the
implementation of changes together with the TropSOC core team. Within the pool of data
contributors, individuals with significant experience working with TropSOC may be designated,
either by the steering committee or database maintainers, as expert reviewers. These individuals
are tasked to assist maintainers and oversee peer review and quality assessment of contributed
new entries.

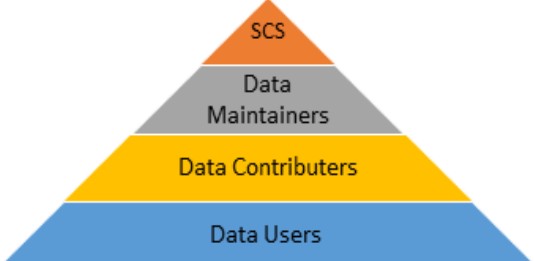


**Figure 7.** A simplified depiction of the TropSOC governance. The scientific steering committee
(SCS) is responsible for approving major management decisions. The TropSOC core team as
data maintainers are responsible for implementing broader changes together with new data
contributors. All interested scientists are welcome to contribute data to future versions of the data
base or access the data for their own research.
**6. Data Availability and User Guidelines**



All data presented in this study is part of the publication and added as a supplement consisting of
datatables (.csv) and accompanying metadata descriptions (.pdf files). In addition, the database
and its metadata is archived and published in the open access environmental and geoscience
data repository at the German Research Centre for Geosciences (GFZ), accessible at:
https://doi.org/10.5880/fidgeo.2021.009. Please note that the database DOI is currently in
preparation and will be released as soon as the review process is completed. In the meanwhile,
please use the following link to access the database (version 1.0) or consult the supplement
added to this submission:
https://dataservices.gfz-
potsdam.de/panmetaworks/review/efed3d5f6035ca261a95aaab45704c2d7d69ac1219d4abd3773
d5f104a4900d3/
Additionally the database is accessible via the website of the Congo Biogeochemistry Repository
(https://www.congo-biogeochem.com/data). Updated versions of the database will be made
available as version updates at both repository.

As detailed above, TropSOC is an open source project that provides several ways for
participation. Anyone may share the TropSOC dataset provided they do so in accordance with
the   Creative   Commons   Attribution   4.0   International   Public   License
(https://creativecommons.org/licenses/by/4.0/legalcode) and by citing the according references
of the original database description and future modifications under their separate DOI.
In addition, we strongly encourage TropSOC users to follow these simple guidelines for use:
(1) TropSOC users must agree not to manipulate the original source data without permission of
the TropSOC governance team described in section 5. This process should be followed in
particular when groups or individuals seek to use the TropSOC database beyond the scope
of its original objectives (see section 1.1).

(2) When utilizing TropSOC data, including the complete dataset, individually curated entries, or
value-added calculations, users should cite this publication and reference the version of
TropSOC that was used for their work under its specific DOI.

When using the database, please cite TropSOC v1.0 as:
Doetterl, S.; Bukombe, B.; Cooper, M.; Kidinda, L.; Muhindo, D.; Reichenbach, M.; Stegmann, A.; Summerauer, L.;
Wilken, F.; Fiener, P. TropSOC Database. Version 1.0. GFZ Data Services. https://doi.org/10.5880/fidgeo.2021.009,
2021.





Additionally, please cite this publication here where the data is first described as:
Doetterl S., Asifiwe R.K., Baert G., Bamba F., Bauters M., Boeckx P., Bukombe B., Cadisch G., Cizungu L.N., Cooper
M., Hoyt A., Kabaseke C., Kalbitz K., Kidinda L., Maier A., Mainka M., Mayrock J., Muhindo D., Mujinya B.B., Mukotanyi,
S.M., Nabahungu L., Reichenbach M., Rewald B., Six J., Stegmann A., Summerauer L., Unseld R., Vanlauwe B., Van
Oost K., Verheyen K., Vogel C., Wilken F., Fiener P. Organic matter cycling along geochemical, geomorphic and
disturbance gradients in forests and cropland of the African Tropics - TropSOC Database Version 1.0. *Earth System
Science* XXX, DOI XXX, 2021.
(3) If users leverage individual data entries from the database, they should also cite the original
research studies in which this particular data has been used for its first time (e.g. Bukombe et
al., 2021, Kidinda et al., 2021; Reichenbach et al., 2021; Summerauer et al., 2021; Wilken et
al., 2021)
(4) When users interpret their own data in the context of data accessed from TropSOC, they
should submit those new data for inclusion in TropSOC after they have published their results
and/or obtained a DOI for their dataset (Details of contributing process see section 5).
**7. Conclusions and Outreach**
The TropSOC database is an attempt to gather the data used in individual studies in one place
and in the same format to facilitate comparisons and synthesis activities. TropSOC is unique in
that it includes measurements and monitoring data of bulk soil and vegetation responses in the
African tropical context for the first time on carefully selected and comparable land use,
geomorphic and geochemical gradients at the landscape scale. Building on the data gathered
along these gradients during several years of field activities and carrying out numerous lab
experiments to investigate the impact of soil geochemistry and land degradation on
biogeochemical cycles in tropical plant-soil systems, TropSOC is the largest integrative project
database on plant-microbial-soil systems in the Congo basin to date. TropSOC's open-access
database structure and participatory approach makes it a suitable tool for scientists to study
experimentally defined soil disturbance and plant responses, as well as to test some of the
assumptions behind modelling biogeochemical cycles in land surface models. Furthermore, we
hope to encourage the community to increase the effectiveness of that investment, and to use the
TropSOC database as a repository to increase the impact of your own research results. As such,
TropSOC is an interactive database that is open for contributions. In addition, TropSOC now
manages one of the largest topically structured soil and plant sample archives for tropical eastern





Africa with several thousand samples and more than three tons of plant and soil material stored at
ETH Zurich. Subsamples of all the above are available upon request to interested researchers.
Finally, we hope that work based on the TropSOC database can help to provide answers on the
role and magnitude of geochemistry, as well as soil mobilization, in controlling biological processes
and fluxes of carbon and nutrients in the Tropics in order to better constrain soil processes in
models ranging from profile to global scales (Todd-Brown et al. 2013). Reducing the uncertainties
associated with our understanding of tropical (agro-) ecosystems in diverse but rapidly changing
landscapes is one of the most pressing issued for securing the future well being of hundreds of
millions of people and to constrain land loss in an area that is home to some of the last and most
fragile populations of great apes in the wild. Elucidating the gravity of the consequences for soil
functioning that can be observed in the TropSOC's study area can contribute to reducing the large
uncertainty associated with terrestrial biogeochemical processes in models and raise awareness
for the necessity of pressing for and creating socio-economic fundament for sustainable land
management in tropical Africa.


## 8. Appendix

**Appendix Table A1.** Basic chemical and physical soil parameters aggregated at land use and geochemical regions. Displayed are average values and standard deviation taken over ten soil increments á 10 cm taken from 0 - 100 cm soil depth derived from NIR-MIR spectral data, calibrated on samples from three depth increments (0 – 10 cm; 30 – 40 cm; 60 – 70 cm). See metadata files 223_soil_spec.pdf and 323_soil_spec.pdf for details. Abbreviations: CEC = potential cation exchange capacity; ECEC = effective cation exchange capacity; Si = Silica; Al = Aluminum; Fe = Iron; Mn = Manganese; SOC = Soil organic carbon; SON = Soil organic nitrogen; P = Phosphorus; TRB = Total reserve in base cations; BD = Bulk density. All assessment methods are explained in the  according .pdf metadata files accompanying the database.

| Geochemical region | Mafic | | Felsic | | Mixed sedimentary rocks | |
|---|---|---|---|---|---|---|
| Land use | Forest n = 169 | Cropland n = 370 | Forest n = 201 | Cropland n = 239 | Forest n = 174 | Cropland n = 305 |
| **Soil Chemistry** | | | | | | |
| pH (KCl) | 3.92 ± 0.45 | 4.21 ± 0.32 | 4.96 ± 0.64 | 5.00 ± 0.44 | 3.48 ± 0.35 | 4.14 ± 0.42 |
| CEC [me/100 g] | 34.14 ± 4.89 | 21.26 ± 7.46 | 15.24 ± 5.37 | 26.33 ± 6.69 | 14.71 ± 11.50 | 19.02 ± 9.17 |
| share of bases in CEC [%] | 13.21 ± 14.16 | 13.90 ± 10.04 | 59.92 ± 20.87 | 52.72 ± 12.75 | 5.66 ± 11.68 | 18.58 ± 17.65 |
| ECEC [me/100g] | 9.12 ± 3.55 | 4.90 ± 3.00 | 10.43 ± 5.40 | 13.74 ± 3.93 | 5.53 ± 2.49 | 6.49 ± 4.63 |
| share of bases in ECEC [%] | 46.08 ± 18.66 | 48.69 ± 15.67 | 81.72 ± 20.67 | 91.74 ± 16.45 | 9.94 ± 15.83 | 41.36 ± 23.13 |
| Si [%] | 12.41 ± 1.36 | 11.88 ± 2.18 | 19.35 ± 2.83 | 16.35 ± 1.88 | 18.99 ± 5.46 | 15.59 ± 1.84 |
| Al [%] | 9.02 ± 1.11 | 6.37 ± 2.39 | 2.81 ± 1.11 | 4.08 ± 1.29 | 3.10 ± 2.92 | 3.20 ± 1.97 |
| Fe [%] | 10.32 ± 1.67 | 10.98 ± 2.58 | 3.50 ± 1.84 | 5.05 ± 1.68 | 5.65 ± 3.54 | 5.77 ± 1.71 |
| Mn [%] | 0.25 ± 0.07 | 0.19 ± 0.10 | 0.14 ± 0.11 | 0.26 ± 0.10 | 0.25 ± 0.09 | 0.08 ± 0.12 |
| SOC [%] | 2.79 ± 1.55 | 2.12 ± 1.24 | 1.17 ± 1.25 | 2.14 ± 1.45 | 2.87 ± 1.82 | 2.49 ± 1.42 |
| SON [%] | 0.28 ± 0.14 | 0.18 ± 0.10 | 0.12 ± 0.12 | 0.22 ± 0.12 | 0.15 ± 0.14 | 0.20 ± 0.12 |
| SOC/SON [-] | 9.09 ± 6.94 | 15.2 ± 7.89 | 12.30 ± 8.78 | 11.67 ± 14.07 | 38.13 ± 46.07 | 20.52 ± 9.07 |
| Total P [%] | 0.20 ± 0.07 | 0.12 ± 0.06 | 0.12 ± 0.06 | 0.30 ± 0.10 | 0.07 ± 0.07 | 0.10 ± 0.08 |
| TRB [%] | 0.56 ± 0.22 | 0.18 ± 0.19 | 0.60 ± 0.27 | 1.03 ± 0.30 | 0.09 ± 0.17 | 0.21 ± 0.30 |
| **Soil Physics** | | | | | | |
| BD [g/cm³] | 1.20 ± 0.14 | 1.28 ± 0.16 | 1.64 ± 0.16 | 1.41 ± 0.16 | 1.43 ± 0.34 | 1.42 ± 0.19 |
| clay [%] | 54.79 ± 11.79 | 64.76 ± 13.00 | 41.45 ± 11.44 | 35.17 ± 11.26 | 39.60 ± 14.77 | 43.12 ± 11.40 |
| silt [%] | 13.94 ± 2.29 | 11.01 ± 3.28 | 10.23 ± 3.70 | 14.42 ± 3.76 | 21.73 ± 13.03 | 14.45 ± 5.20 |
| sand [%] | 31.39 ± 10.20 | 24.84 ± 9.55 | 51.08 ± 10.52 | 48.81 ± 8.11 | 39.10 ± 18.69 | 41.50 ± 9.15 |





**Appendix Table A2.** Structure of the TropSOC database. For each topic a .pdf file is given that
entails an overview for the available data on soil, vegetation and weather data collected for the
investigated forest and cropland plots. Each dataset then comprises a data-containing .csv file
and an additional metadata-containing .pdf file of the same name.

| | Introduction & structure of the data base | 0_intro_structure.pdf |
|---|---|---|
| **1.** | **Basic information** | **1_basic_information.pdf** |
| 1.1. | Location and basic background information for all plots and points where data were collected | 11_plots_points.csv/pdf |
| 1.2. | Data base internal connection between location of plots and points and soil data from different soil depths | 12_sample_identifier.csv/pdf |
| **2.** | **Forest** | 2_forest.pdf |
| 2.1. | Vegetation | |
| 2.1.1. | Forest inventory | 211_forest_invent.csv/pdf |
| 2.1.2. | Forest inventory aggregated | 212_forest_invent_agg.csv/pdf |
| 2.1.3. | Fresh leaves chemistry | 213_fresh_leaves.csv/pdf |
| 2.1.4. | Fresh leaves chemistry aggregated at species level | 214_fresh_leaves_agg.csv/pdf |
| 2.1.5. | Litter fall | 215_litter.csv/pdf |
| 2.1.6. | Litter fall aggregated to seasonal values | 216_litter_seasonal.csv/pdf |
| 2.1.7. | Litter fall aggregated to annual values | 217_litter_annual.csv/pdf |
| 2.1.8. | Root biomass | 218_root_biomass.csv/pdf |
| 2.1.9. | Root biomass aggregated to seasonal values | 219_root_biomass_seasonal.csv/pdf |
| 2.1.10. | Root biomass aggregated to annual values | 2110_root_biomass_annual.csv/pdf |
| 2.1.11. | Root productivity | 2111_root_prod.csv/pdf |
| 2.1.12. | Root productivity aggregated to seasonal values | 2112_root_prod_seasonal.csv/pdf |
| 2.1.13. | Root productivity aggregated to annual values | 2113_root_prod_annual.csv/pdf |
| 2.2. | Mineral soil layers | |
| 2.2.1. | Soil carbon and nitrogen including different organic matter fractions | 221_soil_carbon.csv/pdf |
| 2.2.2. | Physical and chemical soil properties from traditional laboratory analyses. | 222_soil_phy_chem.csv/pdf |
| 2.2.3. | Physicochemical soil properties from NIR-MIR spectroscopy | 224_soil_spec.csv/pdf |
| 2.3. | Organic soil layers | |
| 2.4. | Pu soil inventory | 231_soil_organic_layer.csv/pdf |
| 2.5. | Soil experiments | 241_pu_inventory.csv/pdf |
| 2.5.1. | Incubation experiments | 251_incubation.csv/pdf |
| 2.5.2. | Microbial biomass and enzyme experiments | 252_microbiology.csv/pdf |
| 2.5.3. | $^{14}$C data from bulk soil and $CO_2$ measurements | 253_c14.csv/pdf |
| 2.6. | Parent material | 261_rocks.csv/pdf |
| 2.7. | Soil profile descriptions | 271_profiles.csv/pdf |
| **3.** | **Cropland** | 3_cropland.pdf |
| 3.1. | Biomass & management | |
| 3.1.1. | Biomass yield based on plot data | 311_biomass.csv/pdf |
| 3.1.2. | Land management data | 312_management.csv/pdf |
| 3.2. | Mineral soil layer characterization | |
| 3.2.1. | Soil carbon and nitrogen including different organic matter fractions | 321_soil_carbon.csv/pdf |
| 3.2.2. | Physicochemical soil properties from traditional laboratory methods | 322_soil_phy_chem.csv/pdf |
| 3.2.3. | Physicochemical soil properties from NIR-MIR spectroscopy | 323_soil_spec.csv/pdf |
| 3.3. | $^{239+240}$Pu soil inventory | 331_pu_inventory.csv/pdf |
| 3.4. | Soil experiments | |
| 3.4.1. | Incubation experiments | 341_incubation.csv/pdf |
| 3.4.2. | $^{14}$C data from bulk soil and $CO_2$ measurements | 342_c14.csv/pdf |
| **4.** | **Meteorological data** | 4_meteo.pdf |
| 4.1. | Locations of meteorological stations | 410_meteo_locations.csv/pdf |
| 4.2. | Daily meteorological data from six meteorological stations | 420_meteo_daily.csv/pdf |
| 4.3. | High resolution 5 min triggered precipitation data | 430_meteo_pcp_tig.csv/pdf |
| 4.4. | Meteorological data aggregated to monthly and seasonal values | 440_meteo_monthseas.csv/pdf |





**9. Sample availability**
Remaining soil and plant samples are logged and barcoded at the Department of Environmental
Science at ETH Zurich, Switzerland.
**10. Team list**
See acknowledgements and author list.
**11. Author contribution statement**
SD functioned as the project leader. SD and PF were lead coordinators for compiling the data
base, responsible for data analysis and designed the metadata. BB, MC, LK, DM, MR, LS and FW
were collecting and creating datasets and also analyzed these data before inclusion into the
database. RKA, FB, MC, CB, AM, MM, JM, SMM, LN, AS, RU and CV were technical contributors
and participated via data collection. GB, MB, PB, GC, LNC, AH, KK, BBM, BR, JS, BV, KVO and
KV were conceptual contributors and participated in the design of the study as well as by giving
advice and feedback during the campaign. SD and PF wrote the paper. All authors supported data
analysis and gave feedback during the writing process.
**12. Competing interests**
All other authors declare that they have no conflict of interest.



**13. Special issue statement**
Data presented in this article is the fundament for several research articles published as part of
the Copernicus Special Issue in SOIL with the title: *Tropical biogeochemistry of soils in the Congo*
*Basin and the African Great Lakes region.*
**14. Acknowledgement**

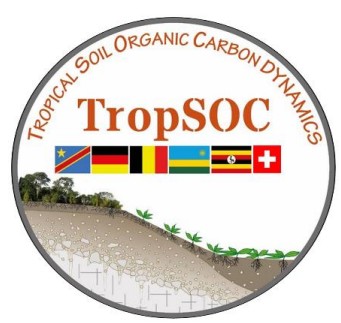

This work is part of the DFG funded Emmy Noether Junior Research Group "Tropical soil organic carbon dynamics along erosional disturbance gradients in relation to soil geochemistry and land use" (TROPSOC; project number 387472333). Micrometeorological data from three of our weather stations (Bukavu, Lukananda, Bugulumiza) were made available and are administered by the Trans-African Hydro-Meteorological Observatory (TAHMO). The authors

like to thank in particular the following collaborating institutions for the support given to our
scientists and this project: International Institute of Tropical Agriculture (CGIAR-IITA), Catholic
University of Bukavu (UCB), Mountain of the Moon University (MMU), Kyaninga Forest Foundation
(KFF), ETH Zurich and the Max Planck Institute for Biogeochemistry in Jena. Special thanks goes
to the many student assistants for their important work in the laboratory and all guards, sentinels
and field work helpers making the sampling campaign possible under difficult conditions.

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

Methods in Establishment Questionnaire Testing: The 2017 Census of Agriculture Testing Bento
Box. Journal of Official Statistics, 34, 341–364, http://dx.doi.org/10.2478/JOS-2018-0016, 2018.
Méchain, M., Ariane, T., Piponiot, C., Chave, J., and Hérault, B.: Biomass: An R Package for
estimating above-ground biomass and its uncertainty in tropical forests, Methods Ecol. Evol., 8,
1163–1167, doi:10.1111/2041-210X.12753, 2017.
Mohr, E.C.J., and van Baren, F.A.: Tropical Soils: A Critical Study of Soil Genesis as Related to
Climate, Rock and Vegetation, 1954.

Mohr, E.C.J., van Baren, F. A. and van Schuylenborgh, J.: Tropical Soils: a comprehensive study
on their genesis, The Hague, 1972.





Nadeu, E., Gobin, A., Fiener, P., Van Wesemael, B. and Van Oost, K.: Modelling the impact of
agricultural management on soil carbon stocks at the regional scale: the role of lateral fluxes,
*Glob. Chang. Biol., 21, 3181-3192,* doi:10.1111/gcb.12889, 2015.
Ohashi, M., Nakano, A., Hirano, Y., Noguchi, K., Ikeno, H., Fukae ,R., Yamase, K., Makita, N.,
and Finer, L.: Applicability of the net sheet method for estimating fine root production in forest
ecosystems. Trees Struct. Funct., 30, 571–578, doi:10.1007/s00468-015-1308-y, 2016.
Ostonen, I., Lõhmus, K., and Pajuste, K.: Fine root biomass, production and its proportion of NPP
in a fertile middle-aged Norway spruce forest: comparison of soil core and ingrowth core methods,
For. Ecol. Manage., 212, 264–277, doi:10.1016/j.foreco.2005.03.064, 2005.
Pan, Y., Birdsey, R.A., Fang, J., Houghton, J.R., Kauppi, P.E., Kurz, W.A., Phillips, O., Shvidenko,
A., Lewis, S.L., Canadell, J.G., Ciais, P., Jackson, R.B., Pacala, S.W., McGuire, D., Piao, S.W.,
Rautiainen, A., Sitch, S., Hayes, D., a D. McGuire, S. Piao, A. Rautiainen, S. Sitch, and Hayes,
D.: A large and persistent carbon sink in the world's forests, Science, 333, 988–993,
doi:10.1126/science.1201609, 2011.
Park, J.H., Meusburger, K., Jang, I., Kang, H., Alewell, C.: Erosion-induced changes in soil
biogeochemical and microbiological properties in Swiss Alpine grasslands. *Soil Biol. Biochem.,*
69, 382-392, doi:10.101/j.soilbio.2013.11.021, 2014.
Pérez-Harguindeguy, N., Díaz, S., Garnier, E., Lavorel, S., Poorter, H., Jaureguiberry, P., Bret-
Harte, M. S., Cornwell, W. K., Craine, J. M., Gurvich, D. E., Urcelay, C., Veneklaas, E. J., Reich,
P. B., Poorter, L., Wright, I. J., Ray, P., Enrico, L., Pausas, J. G., de Vos, A. C., Buchmann, N.,
Funes, G., Quétier, F., Hodgson, J. G., Thompson, K., Morgan, H. D., ter Steege, H., van der
Heijden, M.G. A., Sack, L., Blonder, B., Poschlod, P., Vaieretti, M. V., Conti, G., Staver, A. C.,
Aquino, S., Cornelissen, J. H. C.: New handbook for standardised measurement of plant
functional traits worldwide. Aust. J. Bot., 64, 715–716, doi:10.1071/BT12225_CO, 2016.
Phillips, O., Baker, T., Feldpausch, T. and Brienen, R.: RAINFOR Field Manual for Plot
Establishment and Remeasurement, Pan-Amazonia, Gordon and Betty Moore Foundation, The
Royal Society and European Research Council, 2016.



Reichenbach, M., Fiener, P., Garland, G., Griepentrog, M., Six, J. and Doetterl, S.: The role of
geochemistry in organic carbon stabilization in tropical rainforest soils, Soil Discussion [pre-print],
doi:10.5194/soil-2020-96,  05 January 2021.

Sahani, U. and Behera, N.: Impact of deforestation on soil physicochemical characteristics,
microbial biomass and microbial activity of tropical soil, Land Degr. Develop., 12, 93-105,
doi:10.1002/ldr.429, 2001.

Schimel, D., Pavlick, R., Fisher, J. B., Asner, G. P., Saatchi, S., Townsend, P., Miller, C.,
Frankenberg, C., Hibbard, K. and Cox, P.: Observing terrestrial ecosystems and the carbon cycle
from space, Glob. Chang. Biol., 21, 1762–1776, doi:10.1111/gcb.12822, 2015.

Schlüter, T., and Trauth, M. H.: Geological atlas of Africa: with notes on stratigraphy, tectonics,
economic geology, geohazards and geosites of each country, Springer, Berlin, New York, 272
pp., 2006.

Ssali, H., Ahn, P.M. and Mokwunye, A. U.: Fertility of soils of tropical Africa: A historical
perspective In: Mokwunye AU and Vlek PLG (eds.) Management of Nitrogen and Phosphorus
Fertilizers in sub-Saharan Africa. Martinus Nijhoff, Dodrecht, The Netherlands, 1986.

Summerauer, L., Baumann, P., Ramires-Lopez, L., Barthel, M., Bukombe, B., Reichenbach, M.,
Boeckx, P., Kearsely, E., Van Oost, K., Vanlauwe, B., Chiragaga, D., Heri-Kazi, A.B., Moonen,
P., Sila, A., Shepherd, K., Bazirake, Mujinya, B., Van Ranst, E., Baert, G., Doetterl, S. and Six,
J.: Filling a key gap: a soil infrared library for central Africa, Soil Discussion [pre-print],
doi:10.51947soil-2020-99, 08 January 2021.

Tang, J. and Riley, W.J.: Weaker soil carbon-climate feedbacks resulting from microbial and
abiotic interactions, *Nat. Clim. Chang.,* 5, 56-60, doi:10.1038/nclimate2438, 2015.

Todd-Brown K.E.O., Randerson, J.T., Post, W.M., Hoffman, F.M.,Tarnocai, C., Schuur, E.A.G.
and Allison, S.D.: Causes of variation in soil carbon simulations from CMIP5 Earth system models
and comparison with observations. Biogeosciences 1, 1717–1736, doi:10.5194/bg-10-1717-
1101  2013, 2013.




Tyukavina, A., Hansen, M., Potapov, P., Parker, D., Okpa, C., Stehman, S.V., Kommareddy, I.
and Turubanova, S.: Congo Basin forest loss dominated by increasing smallholder clearing,
Science Advances, 4, doi:10.1126/sciadv.aat2993, 2018.

UN World Heritage Centre. 2010. World Heritage in the Congo Basin. UNESCO, Paris, France.
United Nations Educational, Scientific and Cultural Organization (UNESCO) and World Heritage
Centre (WHC): World Heritage in the Congo Basin, Paris, France,
https://whc.unesco.org/document/104482, 2010.

van Breugel, P., Kindt, R., Lillesø, J-P.B., Bingham, M., Demissew, S., Dudley, C., Friis, I.,
Gachathi, F., Kalema, J., Mbago, F.M.: Potential Natural Vegetation Map of Eastern Africa
(Burundi, Ethiopia, Kenya, Malawi, Rwanda, Tanzania, Uganda and Zambia). Version 2.0, 2015.

Veldkamp, E., Schmidt, M., Powers J.S. and Corre, M.D.: Deforestation and reforestation impacts
on soils in the tropics, Nat. Rev. Earth Environ., 1, 590-605, doi:10.1038/s43017-020-0091-5,
1118 2020.


Verhegghen, A., Mayaux, P., de Wasseige, C., and Defourny, P.: Mapping Congo Basin
vegetation types from 300 m and 1 km multi-sensor time series for carbon stocks and forest areas
estimation, Biogeosciences, 9, 5061–5079, https://doi.org/10.5194/bg-9-5061-2012, 2012.

Vitousek, P.M.: Litterfall, nutrient cycling, and nutrient limitation in tropical forests, Ecology, 65,
1125 285-298,1984.


Walker, T.W. and Syers, J.K.: The fate of phosphorus during pedogenesis, Geoderma, 15, 1-19,
doi:10.1016/0016-7061(76)90066-5, 1976.

Wilken, F., Fiener, P., Ketterer, M., Meusburger, K., Muhindo, D.I., van Oost, K. and Doetterl, S.:
Assessing soil erosion of forest and cropland sites in wet tropical Africa using $^{239+240}$Pu fallout
radionuclides, SOIL [pre-print], doi:10.5194/soil-2020-95, 28 December 2020.

International Union of Soil Sciences (IUSS) Working Group WRB: World reference base for soil
resources 2014. Update 2015, World Soil Resources Reports No. 106, FAO, Rome, 2014.




Zanne, A.E., Lopez-Gonzalez, G., Coomes, D.A., Ilic, J., Jansen, S., Lewis, S.L., Miller, R.B.,
Swenson, N.G., Wiemann, M.C. and Chave, J.: Global wood density database. Dryad Digital
Repository, http://datadryad.org/handle/10255/dryad.235, 2009.