# Peer review of "Organic matter cycling along geochemical, geomorphic and disturbance gradients in forests and cropland of the African Tropics - Project TropSOC Database Version 1.0"

_Earth System Science Data, 2021_

## Author Response (AR1)

Dear editors, dear reviewers,

Many thanks for your helpful and insightful comments on our manuscript draft "ESSD-2021-73" **"Organic matter cycling along geochemical, geomorphic and disturbance gradients in forests and cropland of the African Tropics – Project TropSOC Database Version 1.0"** by Doetterl et al. In the following we would like to give a brief answer the questions and comments you raised and also how we modified the manuscript and database in response to your feedback. Please note that the updated and revised database is ready for submission to the GFZ repository. We are happy to upload it for the case that you are happy with our revisions to it and as soon as we have answered all comments to your satisfaction.  Below our detailed point-by-point response to the reviewer comments (original reviewer comment in italic). Please note that line numbers refer to the final, revised version. An additional Track-Changes version of the manuscript is also provided.

**Reviewer 1  (Dr Yao Zhang)**

**General Assessment**

*"The authors described the open-access database TropSOC 1.0. It is very well written and provided enough details about the data, its experimental design, rationale, references for protocols, and availability of the data. I am very excited to see this database is available for the research community and appreciate the effort of the authors and other supporting personnel. As the authors stated that such data are rare for African tropical regions, I believe this data set is very valuable. I think the manuscript can be accepted with only a few minor changes. "*

**Our response:** Many thanks for the review and we are happy to see that we could satisfy your expectations for the quality of the submitted manuscript and data and are thankful for acknowledging the potential of our database.

**Specific comments**

1. *"It is better to have the full names for CGIAR-IITA and CGIAR-ICRAF"*

**Our response:** Thank you. We will spell out the full name of the two CGIAR branches in full upon first appearance in the manuscript in line 663-664.

2. *"The values in the csv files in the database are semi-colon delimited instead of using commas.  Is there any reason? The standard csv files use comma and can be populated into columns when they are opened by Microsoft Excel."*

**Our response:** We decided to choose semi-colons since some languages use commas to indicate also the start of decimal positions instead of the decimal point. Since semi-colons are not used in such a conflicting way, we decided to stick to those to allow our data to be sorted in columns.

3. *"I found the dates of the sampling are not in the database. For example, the harvest date of biomass. I believe the date is important for some potential users of the database. Maybe these data can be added in the next version of the database."*

**Our response:** We have revisited and amended all data files that did not contain the data of sampling/harvesting. Specifically, the missing information was added to the following files and the database at the GFZ repository updated accordingly: 12_sample_identifier.csv; 311_biomass.csv; 218_root_biomass.csv; 241_pu_inventory.csv; 331_pu_inventory.csv

4. *"In the "Sample availability" section, the authors mentioned the sample of soils and biomass are stored but did not indicate if they are available for researchers outside the group and the policy for sharing these samples. "*

**Our response:** Thanks for pointing this out to us. We have added the following sentences to this section: "As long as idle sample mass remains available, samples for independent research and to stimulate collaboration with the CBO network and the TropSOC project group are available upon request. The authors cannot guarantee to revisit the study sites in order to provide additional sample material in future campaigns. Samples will be given to researchers free of charge. Sample preparation and transport are subject to a handling fee." (line 831-836).

**Reviewer 2 (Dr Rose Abramoff)**

**General Assessment**

*"This ESSD manuscript provides a complete description of the TropSOC V1.0 database, which is to my knowledge the most detailed collection of measurements focusing on tropical central Africa. Measurement methods are well described and cited, and the database structure is also clearly explained. Many of these measurements would be directly useful to researchers working in the region, or who would conduct larger-scale synthesis or modeling studies. The data files are well structured with good meta-data and therefore I would recommend ESSD to publish this manuscript."*

**Our response:** Many thanks for the review and we are happy to see that we could satisfy your expectations for the quality of the submitted manuscript and data and are thankful for acknowledging the potential of our database. We are delighted to hear that you think that our data is useful for large scale modeling or synthesis efforts since this was one of our core intentions with the publication of this data.

*"If I had one recommendation for future versions of TropSOC, it would be that elemental analysis of plant tissues would be useful to modelers as a complement to the growth and biomass measurements, such as the %C and N of litter and root biomass, for example. The elemental analysis of soil and parent material is impressive, and I understand that soils are the main focus of the current study."*

**Our response:** Thank you very much for this comment. We are actually working to complete CN data from vegetation samples right now. Due to the large volume and workload related to this (our litter data was collected every two weeks, root data seasonally) we have not completed this task yet and are delayed due to Covid-19 related restrictions on the ground in our partner institutions, but also in our own laboratories. Once the situation improved, we will create an updated version of our database where we will add this data as 2 new files "224_VegetationCN.csv" and "334_VegetationCN.csv" to the database in the GFZ repository. In these files we will report averaged values (at the plot and sample ID level) for litter and fine root samples (which drive the C input in soils), aggregated to a seasonal time resolution. Note that for forest sites, CN values from sampled fresh canopy leaves are already available in files "213_fresh_leaves.csv" and "214_fresh_leaves_agg.csv" in the current version.

**Specific comments**

1. *L121-122: Seemed like you meant to say "a decrease in biomass productivity". The second half of this sentence is a little hard to understand – negative like lower activity?*

**Our response:** We amended the sentence in the following way: **…**plants leading to a decrease in biomass productivity (Veldkamp et al. 2020) and degraded tropical forests, lowering also microbial activity in soils (Sahani & Behera, 2001).

2. *L159-163: This sentence is a bit unwieldy – can it be condensed or broken into two sentences?*

**Our response:** We amended the sentence in the following way: **"**Improving our process understanding on the coupling between soil biogeochemistry and plant responses in the context of tropical land use changes will help to better constrain plant-soil interactions in ecosystem and land surface models. **Furthermore, insights in plant- soil interactions can help** to better inform policy makers and stakeholders in improving land management practices."

3. *L280: Typo "Overview of plots"*

**Our response:** Corrected, thanks!

4. *L496: Typo "were used for yet more detailed"*

**Our response:** Corrected, thanks!

5. *Typo in PDF of "252_microbiology.pdf" (the .csv is all good): plot_ID and sample_ID should be plotID and sampleID*

**Our response:** Corrected, thanks!

We hope we could address all comments to your satisfaction. Once again, thank you for the helpful reviews.

Yours sincerely,

the authors.